# A Generalized Dynamic Model and Coupling Meshing Force Analysis for Planetary Gear Set Transmissions

Haiwei Wang [1,*], Cheng Ji [1], Fengxia Lu [2], Cheng Wang [3] and Xueyan Sun [3]

1 Shaanxi Engineering Laboratory for Transmissions and Controls, Northwestern Polytechnical University, Xi'an 710072, China; zacharychelsea@outlook.com
2 National Key Laboratory of Science and Technology on Helicopter Transmission, Nanjing University of Aeronautics and Astronautics, Nanjing 210016, China; meefxlu@nuaa.edu.cn
3 Key Laboratory of Vehicle Transmission, China North Vehicle Research Institute, Beijing 100072, China; bitstudentwc@163.com (C.W.); johnsxy1993@sina.cn (X.S.)
* Correspondence: whw@nwpu.edu.cn

**Abstract:** The dynamics analysis of a planetary gear set transmissions requires the creation of completely different models for different gears, which is very tedious. In this paper, a generalized dynamics modeling process is proposed for a three planetary gear set transmissions, and a generalized dynamic model for multiple gears is established by using the lumped mass method. The analysis of meshing force characteristics is carried out for the second gear position, and the meshing frequency coupling phenomenon between the meshing forces of the three planetary gear sets is investigated. The results show that, for the current gear set of meshing force, the meshing frequency components of other gear sets only appear in a part of the speed, and with the increase in speed, certain low-frequency components of other sets that exist at low speed will decrease or even disappear, and the coupling relationship between the meshing forces of different planetary gear sets is not symmetrical.

**Keywords:** planetary gear set transmissions; generalized dynamic model; dynamics analysis; meshing force

## 1. Introduction

The planetary gear drive system is a key component in the process of vehicle transmission. It mainly consists of a gear meshing pair, a transmission connecting shaft, bearing supporting parts and various control components. Compared with the traditional fixed-shaft gear transmission system, the planetary gear transmission system is more compact in structure, smooth in transmission, and higher power transmission and wider range of transmission ratio can be achieved. The planetary row gear transmission system is the key device in transmissions; its performance directly affects system performance. The device is mostly a coupling mechanism. Aiming at the different connection forms of the structure between different gears, the study on the generalization method suitable for different gears and the rapid composition of the dynamic equation of the whole mechanism can provide convenience for the subsequent dynamic characteristics analysis.

For many years, the research of multistage planetary gear transmission system is mainly based on simple planetary gear system dynamics model. Kahraman [1] found that as long as the lateral support stiffness of the mechanism is greater than 10 times the ratio of meshing stiffness, the pure torsional model could be used to calculate the inherent characteristics, and the results obtained by the translation–torsional coupling model are highly consistent. Lin [2] proposed a translation–torsional coupling dynamic model of planetary gears by taking translational degrees of freedom into account for all components and studied their natural frequencies and vibration modes. Bang [3] proposed a multi-speed transmission mechanism that uses a compound planetary gear set and brakes. Ambarisha [4] proposed a lumped mass model and a two-dimensional

finite element contact model for the spur planetary gear system to study the non-linear dynamic behaviors of the system. Zhao [5] established a flexible model of planetary gear system based on finite element theory, simulated the actual working conditions for simulation analysis of the inherent properties of the system, and predicted the dangerous parts of the meshing pair. Lai [6] proposed a flexible–rigid coupling dynamic (FRCD) model by coupling the condensed substructure of finite element (FE) flexible bodies with rigid dynamic model and investigated the effect of sun gear positions on the dynamic response of the long planet and the mechanism for the vibration of the floating ring gear. Benford [7], Kim [8], Kahraman [9], Lang [10] and Tian [11] carried out statics and kinematics analysis on gear components of planetary automatic transmission. For statics analysis of compound planetary gear sets, Kahraman [9] proposed a generalized formula for speed and force analysis of gear parts of planetary automatic transmission, which has certain applicability to compound planetary gear sets. In addition, researchers use graph theory to assist in kinematic and statical analysis. Lang [10] used graph theory to analyze the motion and torque of fixed gear train and epicyclic gear train. Tian [11] used the establishment of the corresponding matrix to assist in completion of the planetary transmission motion and torque analysis. For the dynamics modeling method, based on the idea of generalized finite element method (fem) and the system discretization subunits of lumped mass model, Chang [12] proposed the subunit assembly of the coupling system dynamic model of the general modeling method. Based on this method, a general dynamic modeling method considering gears shift is proposed in this thesis. Qian [13] established translation–torsional coupling dynamics model of a Lavina-type composite planetary array under multiple working conditions in the Matlab environment. Jing [14] presented a fully coupled dynamic modeling strategy for a planetary gear system. Lv [15] established a 12-degree-of-freedom spur gear dynamic model, which is coupled by the mesh gear pair and the gearbox. Xiao [16] studied the vibration characteristics and meshing force spectrum components of the three-stage planetary gear system. Liu [17] derived the translational and torsional coupling dynamics model of two-stage planetary gear, studied the connection between inter-stage connecting shaft and frequency coupling of two-stage meshing force, and analyzed the dynamic characteristics of different gear sets. Wei [18] established shaft elements of different components in consideration of component flexibility and predicted dynamic vibration behavior of multistage planetary gear system. Zhang [19] modified the dynamic parameters of the inherent characteristics of the three-planet array based on parameter sensitivity analysis, effectively avoiding torsional vibration. Dou [20] studied the main frequency components of gear meshing force and spindle shear force of a composite planetary transmission system at variable speed. Hao [19] took the two-stage planetary gear system as a research object to analyze the influence of parallel and series relations between multiple gear sets on the coupling components of meshing forces.

According to the different methods and factors considered in the establishment of a system coupling dynamics model, the established dynamics model mainly includes the following types: lumped mass model, finite element model and lumped mass-finite element mixed model. However, the above modeling method can only establish the structure of components in a specific combination form. Aiming at the relationship between dynamic models established by different gears, the concept of gear matrix of dynamic parameter is introduced and relevant rules are formulated. A general dynamic model building method considering gears' transformation is proposed. Thus, the dynamic model of the whole mechanism can be formed quickly, which is convenient for the subsequent dynamic characteristic analysis. In this paper, a translation–torsional dynamics model of a three planetary gear set transmissions is established based on the lumped mass method. On this basis, the concept of gear matrix of dynamic parameters is introduced, relevant rules are formulated, and a generalized dynamic model construction method considering gear status is proposed. A Fourier series method is used to calculate the dynamic meshing force, and the spectrum characteristics of the external meshing force of three planetary gear set

transmissions are analyzed under the condition of variable speed, and the existence forms of meshing frequency coupling phenomena are studied.

## 2. Generalized Dynamic Model

The object of this paper is a three planetary gear set transmissions, which contains six shift control components and three planetary gear sets of 2K-H straight planetary gear sets, as shown in Figure 1. Among them, the shift control element consists of three clutches (CL1, CL2, CL3) and three brakes (B1, B2, B3), a straight-gear planetary gear set of Pi (i = 1, 2, 3) which includes sun gear, ring gear, and carrier (Si, Ri, Ci) and Ni (N1 = N2 = 4, N3 = 6) planetary gears in each gear set.

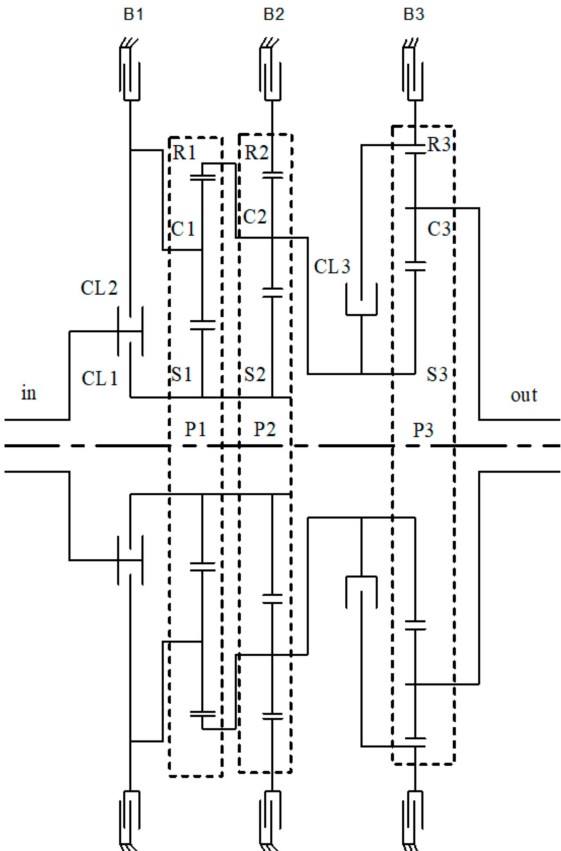

**Figure 1.** Three-planet gear drive system.

### 2.1. Establishment of Global Dynamics Model

The whole system is discretized by using the idea of finite element, and the whole planetary gear system is discretized into sub-elements such as shaft element, external meshing element, internal meshing element, planetary carrier element and bearing element. To be specific, for avoiding unnecessary complexity, only a simplified matrix form of all elements' motion differential equations is shown.

### 2.1.1. Shaft Element

The shaft element is simulated and connected by the lumped mass method, and the force of the shaft element is analyzed, and the dynamic model of the shaft element is established, as shown in Figure 2.

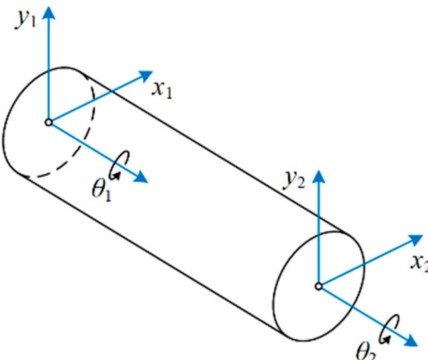

**Figure 2.** Shaft element.

Two mass nodes are connected by one shaft element. A shaft element with two nodes and six degrees of freedom is established between two mass nodes. The generalized coordinate system of the shaft element is assumed as Equation (1).

$$x_s = \{x_1, y_1, \theta_1, x_2, y_2, \theta_2\}^T \tag{1}$$

where $x_1$, $y_1$, $x_2$ and $y_2$ represent the transverse displacements of each node along each coordinate axis, respectively; $\theta_1$ and $\theta_2$, represent the torsional displacements of two mass nodes along their respective torsional directions.

The matrix form of the motion differential equation is shown in Equation (2).

$$M_s \ddot{x}_s + C_s \dot{x}_s + K_s x_s = 0 \tag{2}$$

where $M_s$, $C_s$, and $K_s$ represent the element fixed mass matrix, damping matrix and stiffness matrix, respectively.

### 2.1.2. External Meshing Element

Taking the spring compression direction of the active and passive gears along the meshing line as positive, the vibration displacement and torsional displacement of the three degrees of freedom of the external meshing gear are projected to the meshing line direction, as shown in Figure 3. The relative total deformation of the projection of the gear pair along the meshing line direction $\delta_0$ can be obtained as shown in Equation (3).

$$\delta_0 = V_0 q_0 - e_0 \tag{3}$$

where $q_0 = \{x_s, y_s, \theta_s, x_{pi}, y_{pi}, \theta_{pi}\}$ represents the displacement column vectors of the driving gear node and the slave gear node in the external meshing element, respectively, in the global coordinate system. $e_0$ represents the composite meshing error of external meshing gear pair along the normal direction. $V_0$ is the projection vector of the transformation from the displacement of the degree of freedom in each direction of the external meshing gear pair to the direction of the meshing line, and its expression is shown in Equation (4).

$$V_o = \left[ \sin \psi_{sp}, \cos \psi_{sp}, r_{bs}, -\sin \alpha_{spi}, -\cos \alpha_{spi}, -r_{bp} \right] \tag{4}$$

where $r_{bs}$, $r_{bp}$ represent the base circle radius of the sun gear and the planetary gear, respectively; $\alpha_{spi}$ represents the meshing angle between the i-th planetary gear and the sun gear; $\psi_{spi}$ represents the angle between the meshing line between the driving gear and the i-th planetary gear and the Y-axis, the change in the driving gear's steering will affect $\psi_{spi}$, when the driving gear's rotation is counter clockwise, the sign is positive, and vice versa. The specific expression can be expressed in Equation (5).

$$\psi_{spi} = \alpha_{spi} \pm \varphi_{pi} \tag{5}$$

where $\psi_{spi}$ represents the engagement angle between the i-th planet gear and the driving gear, and $\varphi_{pi}$ represents the positive included angle between the central line vector of the i-th planetary gear and the driving gear and the *X*-axis of the global coordinate system; that is, the installation phase angle.

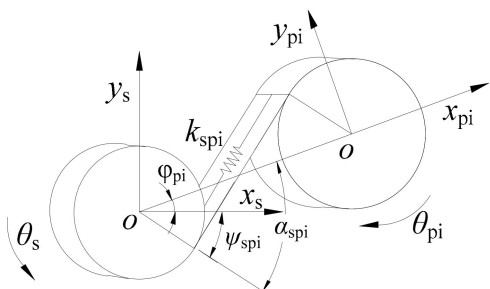

**Figure 3.** External meshing element.

According to Newton's second law, the motion differential equation of the external meshing element can be obtained in the matrix form as Equation (6).

$$M_o\ddot{q}_o + Co(\dot{q}_o - \dot{e}_o) + K_o(q_o - e_o) = F_o \tag{6}$$

where $M_o$ is the mass matrix of the external meshing element, $C_o$ is the damping matrix of the external meshing element, $K_o$ is the stiffness matrix of the external meshing element, $e_o$ and its differential coefficient are the equivalent displacement column vector and equivalent velocity vector of the external meshing element after the integrated meshing error is decomposed in the degrees of freedom in each direction, and $F_o$ is the column vector formed by the components of the excitation force on each degree of freedom.

2.1.3. Internal Meshing Element

Taking the spring compression direction of the driving gear along the meshing line as positive, the vibration and torsional displacements of the three degrees of freedom of each gear are projected to the meshing line, and the dynamic model diagram of the freedom of the inner gear ring and the planetary gear in the translation and torsion direction is obtained, as shown in Figure 4.

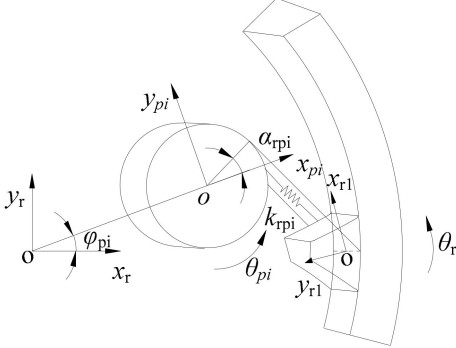

**Figure 4.** Internal meshing element.

The vibration displacement of the internal meshing element in Figure 4 is projected to the meshing line along the degrees of freedom in each direction; that is, the overall deformation between the driven gear and the driving gear along the meshing line direction can be obtained as Equation (7).

$$\delta_{rpi} = V_i q_{rpi} - e_{rpi} \tag{7}$$

where $q_{rpi} = \{x_{pi}, y_{pi}, \theta_{pi}, x_r, y_r, \theta_r\}^T$ is the displacement column vectors of the driving gear node and the driven gear node in the internal meshing element in the global coordinate system, respectively, $e_{rpi}$ is the composite meshing error of the internal meshing gear pair along the normal direction, and $V_i$ is the projection vector of the transformation from the displacement of the freedom of the i-th planetary gear and ring gear in each direction to the meshing line direction, which can be expressed as Equation (8).

$$V_i = \left[\sin \alpha_{rpi}, -\cos \alpha_{rpi}, -r_{bpi}, -\sin \psi_{rpi}, \cos \psi_{rpi}, r_{bri}\right] \tag{8}$$

where $r_{bpi}$ and $r_{bri}$ are the radius of the base circle of the planetary gear and inner ring gear, respectively, $\psi_{rpi}$ is the positive angle between the end meshing line between the i-th planetary gear and the inner ring gear and the Y-axis of the global coordinate system as Equation (9).

$$\psi_{rpi} = \alpha_{rpi} \pm \varphi_{pi} \tag{9}$$

where $\alpha_{rpi}$ is the meshing angle between the i-th planetary gear and the inner ring gear, $\varphi_{pi}$ is the positive included angle between the line vector of the i-th planetary gear and the center of the inner gear ring and the X-axis of the global coordinate system. The value of $\psi_{rpi}$ is affected by the installation phase angle $\varphi_{pi}$. When the rotation direction of the driving gear is counterclockwise, the sign is positive, and vice versa.

The motion differential equation of the internal meshing element can be obtained in the matrix form as Equation (10).

$$M_I \ddot{q}_{rpi} + C_I(\dot{q}_{rpi} - \dot{e}_{rpi}) + K_I(q_{rpi} - e_{rpi}) = F_s \tag{10}$$

where $M_I$ is the mass matrix of the external meshing element, $C_I$ is the damping matrix of the external meshing element, $K_I$ is the stiffness matrix of the external meshing element, $e_{rpi}$, and its differential coefficient are the equivalent displacement column vector and equivalent velocity vector of the integrated meshing error decomposed in the degrees of freedom in each direction for the external meshing element, $F_s$ is the column vector formed by the components of the excitation force on each degree of freedom.

### 2.1.4. Planetary Carrier Element

As a very important component, the planetary carrier not only plays the role of supporting the planetary gear, but also serves as the output component of the system in the third gear set and is the input component of the system in the second and fourth gears. In addition, the planetary carrier in the second gear set serves as a coupling component in the middle of the first gear set and the third set. Figure 5 shows the general planetary carrier element.

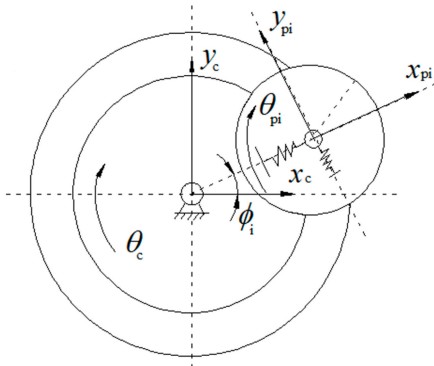

**Figure 5.** Planetary carrier element.

The deformation of planet carrier $\delta_c$ is expressed as Equation (11).

$$\delta_c = V_c q_c \tag{11}$$

where $q_c = \{x_c, y_c, \theta_c, x_{pi}, y_p, \theta_{pi}\}$ represents the displacement column vectors of each node of the planetary carrier element, $V_c$ is the projection vector of the displacements in all directions in the global coordinate system of the planetary carrier, which can be expressed by Equations (12) and (13):

$$V_c^x = [\cos\phi_i, \sin\phi_i, 0, -1, 0, 0] \tag{12}$$

$$V_c^y = [-\sin\phi_i, \cos\phi_i, r_c, 0, -1, 0] \tag{13}$$

where $r_c$ is the radius of the planetary carrier, $\phi_i$ is the mounting position angle of the i-th planetary gaer. For the planetary transmission system with uniform distribution of planetary gears, $\phi_i = 2\pi(i-1)/n$, where, i = 1, 2, ... , n, n is the number of planetary gears.

The motion differential equation of the planetary carrier element can be obtained in the matrix form as Equation (14).

$$M_c\ddot{q}_c + K_c q_c + C_c\dot{q}_c = F_c \tag{14}$$

where $M_c$, $C_c$ and $K_c$, represent mass matrix, damping matrix and stiffness matrix, respectively, $F_c$ is the column vector of the force applied to the planetary carrier.

The system dynamic model diagram by discrete element matrix is shown in Figure 6, in which B1–B3 are three brakers. $K_{sjp}$ and $K_{rjp}$ (j = 1, 2, 3) represent the external and internal meshing stiffness of the three planetary gear sets, respectively. The 25 3-dof mass nodes include the input shaft node in, the coupling disc node LP, the sun gear node Si in i-th set (i = 1, 2, 3), the inner ring gear node Ri, the planetary carrier node Ci, and the planetary gear node $p_{ij}$ in the i-th set.

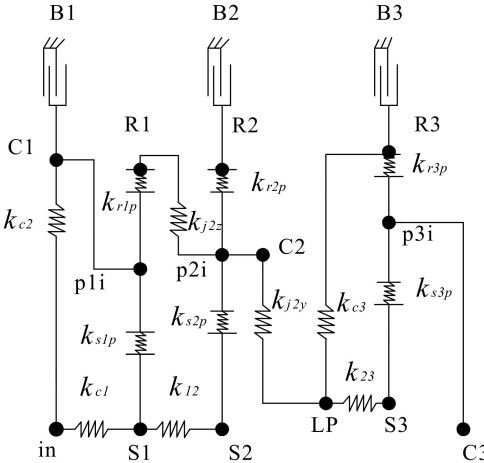

**Figure 6.** Dynamics model of three planetary gear set transmissions.

The meanings of each connection stiffness in Figure 6 are shown in Table 1. The matrix form of the connection stiffness is $diag(k_x, k_y, k_\theta)$, $k_x$, $k_y$ are the bending stiffnesses, and $k_\theta$ is the torsional stiffness.

The dynamics models of the first and second planetary gear set and the third planetary gear set in the dynamics model of the three planetary gear set transmissions are established in two coordinate systems, respectively, as shown in Figure 7a,b. XY in the two figures is the fixed coordinate system of the center gear, and its coordinate origin is located at the theoretical center of the center gear. Horizontally to the right is the positive direction of X, and the positive direction of X rotated 90° counterclockwise is the positive direction of Y. $x_i y_i$ (i = 1, 2, ... , N, N is the number of planetary gears) in two figures for each gear set of planetary gear in the local coordinate system. With the speed of the planet carrier uniform rotation for the moving coordinate system, the origin of the theory of center in the case of a planet gear, center gear's theory and the theory of the planet gear center outside the

connection point to the direction of the positive direction for $x_i$, the positive direction of $x_i$ rotated by 90 degrees counterclockwise is the positive direction of $y_i$.

**Table 1.** The meanings of each connection stiffness in Figure 6.

| Stiffness Symbol | The Meaning of Stiffness Symbol |
|---|---|
| $k_{c1}$ | Connection stiffness between input shaft and S1 when clutch CL1 engages |
| $k_{c2}$ | Connection stiffness between input shaft and C1 when clutch CL2 engages |
| $k_{12}$ | Connection stiffness between sun gear S1 and sun gear S1 |
| $k_{j2z}$ | Connection stiffness between ring gear R1 and planetary carrier C2 |
| $k_{j2y}$ | Connection stiffness between planetary carrier C2 and connecting gear |
| $k_{23}$ | Connection stiffness between connecting gear and sun gear S3 |
| $k_{c3}$ | Connection stiffness between connecting disc and ring gear R3 |

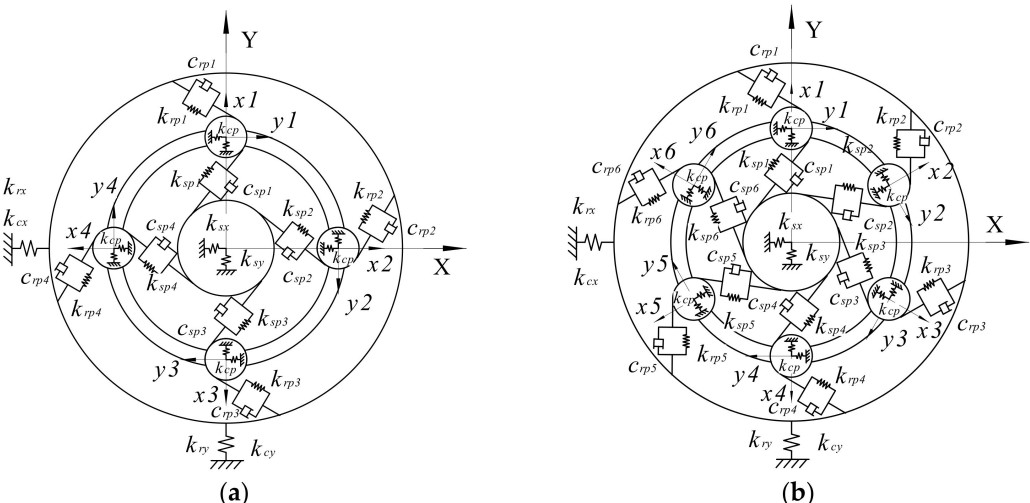

**Figure 7.** Dynamics model of planetary gear sets. (**a**) Dynamics model of the planetary gear set 1 or 2; (**b**) dynamics model of the planetary gear set 3.

The spring supporting elements $k_{jx}$, $k_{jy}$ ($j$ = s, r, c) represent the supporting stiffness of the parts (sun gear, inner gear ring, and planetary carrier, respectively) in planetary gear set along the local coordinate system $x_iy_i$. The spring supporting element $k_{cp}$ represents the supporting stiffness of the three planetary gear sets of respective planetary frames to N planet gears. Spring element $k_{spi}$, $k_{rpi}$ ($I$ = 1, 2, ... , N, N is the number of planet gears), respectively, represent the meshing relationship between the three gear sets of respective sun gears and the i-th planetary gear, and the meshing relationship between the three gear sets of respective inner ring gears of the sun gears and the planetary gear. $c_{spi}$, $c_{rpi}$ are the corresponding meshing damping values of $k_{spi}$, $k_{rpi}$ in their respective meshing relations.

The dynamic model of the whole three planetary gear set transmissions is discretized into sub-elements, including a shaft element, a bearing element, an external meshing element, an internal meshing element, a planetary carrier element, a bearing element, and so on, by using the idea of finite element. The motion differential equations of each sub-element are listed by lumped mass method and converted into the mass matrix, damping matrix, and stiffness matrix, respectively. The motion equation of the planetary gear set transmissions as a whole is shown in Equation (15).

$$M\ddot{x}(t) + C[\dot{x}(t) - \dot{e}(t)] + K(t)[x(t) - e(t)] = P_0 \tag{15}$$

where $M$ is the overall mass matrix of the system; $C$ is the damping matrix of the system; $K(t)$ is the stiffness matrix of the system; $x(t)$ is the displacement column vectors of all mass nodes in the system, $P_0$ is the external load column vector of the system.

## 2.2. A Generalized Dynamic Model for Planetary Gear Set Transmissions

The planetary gear set transmissions has seven gears movement relationship, including five forwards gears and two backwards gears. The state table of control components for all gears is shown in Table 2.

**Table 2.** The state table of control components for all gears *.

| Gear | CL1 | CL2 | CL3 | B1 | B2 | B3 |
|---|---|---|---|---|---|---|
| First gear | ○ | × | × | × | ○ | ○ |
| Second gear | × | ○ | × | × | ○ | ○ |
| Third gear | ○ | ○ | × | × | × | ○ |
| Fourth gear | × | ○ | ○ | × | ○ | × |
| Fifth gear | ○ | ○ | ○ | × | × | × |
| First reverse gear | ○ | × | × | ○ | × | ○ |
| Second reverse gear | ○ | × | ○ | ○ | × | × |

* The symbol of '○'means engaging state and the symbol of '×'means separation state in the table.

Considering the different states of control components in different gears, the corresponding dynamic models are different in braking relations and connection relations. In order to improve the modeling efficiency when shifting between different gears, on the basis of the whole dynamic model, the general dynamic model construction method considering gear shift is studied.

The establishment of the generalized dynamic model considering gear shift actually deals with the dynamic parameter matrices (including the mass matrix, damping matrix, and stiffness matrix) involved in the whole dynamic model of the system. The specific generalized dynamics model processing procedure is shown in Figure 8. In view of the different joint states of two control components, there are two ways to deal with the whole dynamic parameter matrix of the system. When the clutch is engaged, the connection relation between nodes should be considered. The node number of the two nodes connected by the clutch can be preset in advance through the shaft element. By using the connection row vector, the connection relation of different nodes caused by different gears is established and reflected in the dynamic parameter matrices. When the brake is engaged, first of all, it is necessary to judge the preset node number of the node being braked and obtain the transformation matrix and dynamic parameter gear matrix. Then, the original dynamic parameter matrix of the system is processed by using the dynamic parameter gear matrix, and the dynamic model of different gear position can be obtained by integrating the clutch connection relation processing. If the gear needs to be replaced again, the above dynamic parameter matrix processing procedure can be repeated.

When processing the coupling relation of clutch, shaft element is mainly used to deal with. A connection row vector $A$ is defined to describe the join relation between two members joined by the clutch when the clutch engages. If one of the two connected components is denoted as 1, the other as −1, and the others as 0, each two connected components will form a row vector containing only 1, −1, 0, and one row vector represents a connection relation. When the clutch CL1 engages, the input shaft is connected to the sun gear in the first gear set. When the clutch CL2 engages, the input shaft is connected with the inner ring gear of the first gear set. When the clutch CL3 engages, the connection disc is connected with the third gear set of inner ring gear. There are 25 nodes according to the input shaft, the sun gear, the planet gears, the planetary carrier, the inner ring gear of the three planetary gear sets, the connection disc, and so on.

According to different connection relations, the corresponding connection row vector $A$ of the three connection modes is as Equation (16).

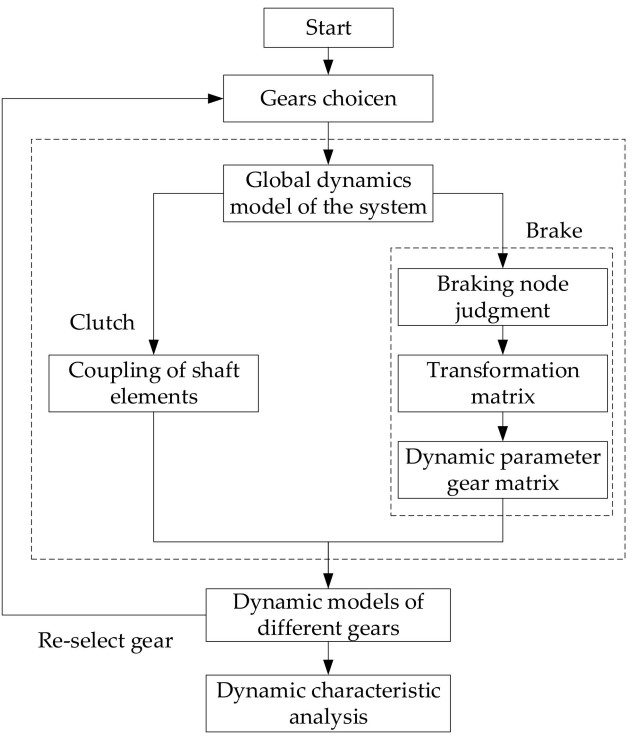

**Figure 8.** Flow chart of generalized dynamics model.

$$A(1) = \begin{bmatrix} 1 & -1 & 0 & \cdots & 0 \end{bmatrix}_{1 \times 25}$$

$$A(2) = \begin{bmatrix} 1 & \underbrace{0 \cdots 0}_{five\,0} & -1 & 0 & \cdots 0 \end{bmatrix}_{1 \times 25}$$

$$A(3) = \begin{bmatrix} \underbrace{0 \cdots 0}_{fifteen\,0} & 1 & 0 & \cdots & 0 & -1 \end{bmatrix}_{1 \times 25} \tag{16}$$

When the clutch engages, the matrix $(A^T A)$ is the direct product of the connection stiffness $K_{ci}$ at the clutch connection, and the obtained matrix is added to the original stiffness matrix to generate the new stiffness matrix considering the clutch connection.

Taking the second gear of the system as an example, the clutch CL2 engages under this gear, and the connection row vector is used to process the stiffness matrix of the system as a whole. The specific matrix processing procedure is as Equation (17).

$$\begin{aligned} A &= A(2), \\ K_{new} &= K + \left( A^T A \right) \otimes K_{c2} \end{aligned} \tag{17}$$

The damping matrix and mass matrix can also be treated using a connection row vector to obtain a new matrix considering the connection relation.

In the process of brake engagement, a transformation matrix is defined, in which the non-braked member is set to one, and each row vector in the matrix represents only one non-braked relation in the whole system.

The 25 mass nodes are sorted according to the previous ordering rules when defining the connection matrix, and seven gear transformation matrices $X_i$ ($i = 1, 2, \ldots, 5, -1, -2$) as Equations (18)–(23). Each row vector in the matrix only represents a non-braking relation in

the whole system, and each row vector represents each mass point according to the above ordering rules from left to right.

$$
X_1 = X_2 = \begin{bmatrix} \begin{pmatrix} 1 & & \\ & \ddots & \\ & & 1 \end{pmatrix}_{14\times14} & \begin{matrix} \vdots \\ 0 \\ \vdots \end{matrix} & & \\ & & \begin{pmatrix} 1 & & \\ & \ddots & \\ & & 1 \end{pmatrix}_{9\times9} & \begin{matrix} \\ 0 \end{matrix} \end{bmatrix}_{23\times25}
\tag{18}
$$

$$
X_3 = \begin{bmatrix} \begin{pmatrix} 1 & & \\ & \ddots & \\ & & 1 \end{pmatrix}_{24\times24} & \begin{matrix} \vdots \\ 0 \end{matrix} \end{bmatrix}_{24\times25}
\tag{19}
$$

$$
X_4 = \begin{bmatrix} \begin{pmatrix} 1 & & \\ & \ddots & \\ & & 1 \end{pmatrix}_{15\times15} & \begin{matrix} \vdots \\ 0 \\ \vdots \end{matrix} & & \\ & & \begin{pmatrix} 1 & & \\ & \ddots & \\ & & 1 \end{pmatrix}_{9\times9} \end{bmatrix}_{24\times25}
\tag{20}
$$

$$
X_5 = \begin{bmatrix} 1 & & & \\ & \ddots & & \\ & & \ddots & \\ & & & 1 \end{bmatrix}_{25\times25}
\tag{21}
$$

$$
X_{-1} = \begin{bmatrix} \begin{pmatrix} 1 & & \\ & \ddots & \\ & & 1 \end{pmatrix}_{6\times6} & \begin{matrix} \vdots \\ 0 \\ \vdots \end{matrix} & & \\ & & \begin{pmatrix} 1 & & \\ & \ddots & \\ & & 1 \end{pmatrix}_{17\times17} & \begin{matrix} \vdots \\ 0 \end{matrix} \end{bmatrix}_{23\times25}
\tag{22}
$$

$$
X_{-2} = \begin{bmatrix} \begin{pmatrix} 1 & & \\ & \ddots & \\ & & 1 \end{pmatrix}_{6\times6} & \begin{matrix} \vdots \\ 0 \\ \vdots \end{matrix} & & \\ & & \begin{pmatrix} 1 & & \\ & \ddots & \\ & & 1 \end{pmatrix}_{18\times18} \end{bmatrix}_{24\times25}
\tag{23}
$$

After the establishment of the shift matrix, since the degree of freedom of each mass point of the system is three, and the direct product of the transition matrix and the unit matrix $I_{3\times3}$ can be successively taken to obtain the dynamic parameter gear matrix $C_{dyn\_i}$ of different gears, as shown in Equation (24).

$$
C_{dyn\_i} = X_i \otimes I_{3\times3}(i = 1, 2, \cdots, 5, -1, -2)
\tag{24}
$$

Similarly, taking the second gear of the system as an example, the processing procedure of dynamic parameter matrix is introduced. The inner ring gear node of No. 15 in the second gear set and the inner gear ring node of No. 25 in the third gear set are braked. There

are 23 non-braking nodes, so there are 23 non-braking relations. The dynamic parameter gear matrix $C_{dyn\_2}$ under this gear is shown in Equation (25).

$$\left(C_{dyn\_2}\right)_{69\times75} = (X_2)_{23\times25} \otimes I_{3\times3} = \begin{bmatrix} \begin{pmatrix} I & & \\ & \ddots & \\ & & I \end{pmatrix}_{(14\times3)\times(14\times3)} & \vdots & O_{3\times3} & \\ & \vdots & & \\ & & \begin{pmatrix} I & & \\ & \ddots & \\ & & I \end{pmatrix}_{(9\times3)\times(9\times3)} & \vdots & O_{3\times3} \end{bmatrix} \tag{25}$$

The new stiffness matrix $(K_{2new})_{69\times69}$, excluding the inner ring node of No. 15 in the second gear set and the inner ring node of No. 25 in the third gear set can be obtained by using the dynamic parameter gear matrix $C_{dyn\_2}$ for gear set transformation and column transformation. The specific calculation process is shown in Equation (26).

$$(K_{2new})_{69\times69} = (C_{dyn\_2})_{69\times75}(K_{new})_{75\times75}(C_{dyn\_2})^T_{75\times69} \tag{26}$$

The damping matrix and mass matrix can also be transformed by using the dynamic parameter gear matrix to generate a new matrix considering the braking relationship.

## 3. Meshing Force Analysis

To analyze the external meshing dynamic force of the first planetary gear in each gear set under the second gear, a Fourier series method is adopted to solve the motion equations of the planetary gear set transmissions. The spectrum analysis under variable speed is carried out to analyze the influence of the meshing force coupling components of each gear set on it and the relationship between them and the input speed. When the input torque $T_{in}$ is 5000 N · m, the dynamic meshing force spectrums of the planetary gear drive system are shown in Figures 9–11 for different speeds.

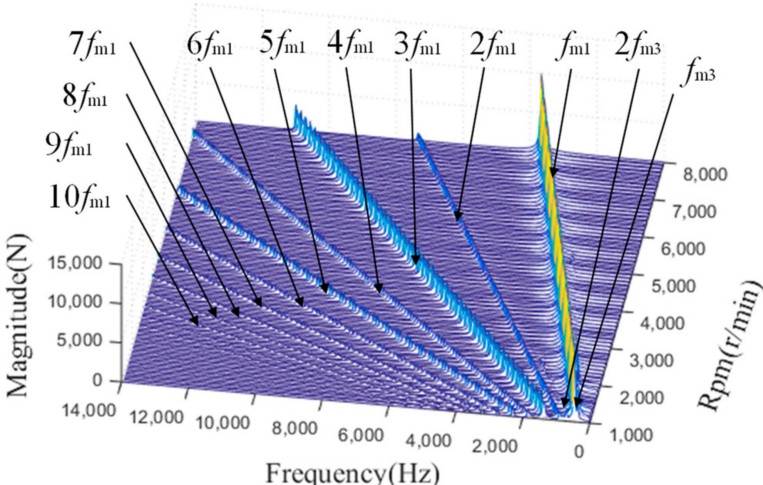

**Figure 9.** External meshing force of the first gear set.

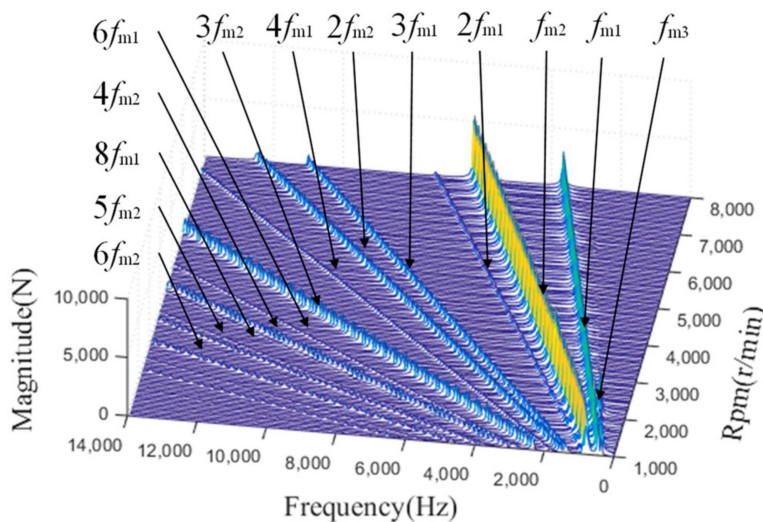

**Figure 10.** External meshing force of the second gear set.

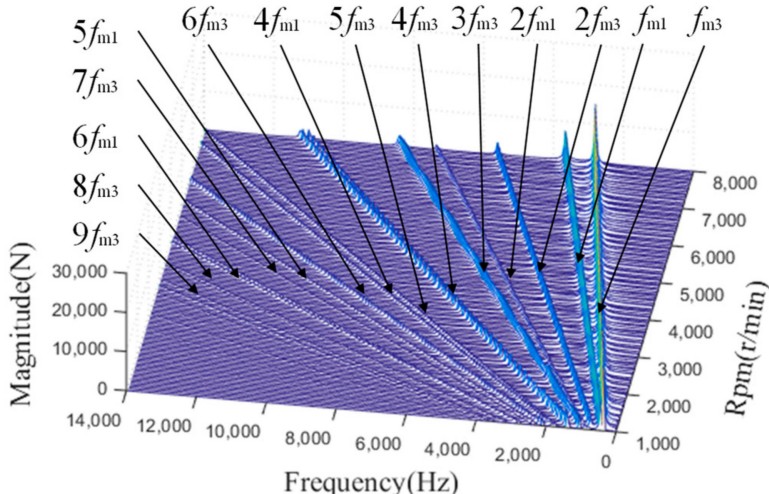

**Figure 11.** External meshing force of the third gear set.

It can be seen from Figures 9–11 that when the rotational speed goes from 1000 r/min to 8000 r/min, the main components of meshing force spectrum in planetary gear set S1–S3 are the meshing frequency $f_{\mathrm{m}i}$ ($i$ = 1, 2, 3) and its frequency doubling of the current set, and the amplitude of frequency doubling component of the current set is the highest. Figure 9 shows that the spectrum of S1 set meshing force mainly consists of the meshing frequency of the first row and its frequency doubling, and the amplitude of the single frequency component is the largest, meshing frequency components $f_{\mathrm{m}3}$ and $2f_{\mathrm{m}3}$ of the S3 set exist in the meshing force spectrum of S1 set at low rotational speed, and these frequency components disappear with the increase in rotational speed. As shown in Figure 10, the spectrum of the S2 set meshing force mainly consists of the meshing frequency of the first row and its frequency doubling, and the amplitude of the single frequency component is the largest. The meshing frequency component $f_{\mathrm{m}3}$ of the gear set S3 exists in the meshing force spectrum of the gear set S2 at low rotational speed, and this frequency component disappears with the increase in rotational speed. In addition, meshing frequency component $f_{\mathrm{m}1}$ of the gear set S1 and its frequency doubling exist, and do not seem to disappear with the increase in rotational speed. As shown in Figure 11, the spectrum of the S3 set meshing force mainly consists of the meshing frequency of the first row and its frequency doubling, and the amplitude of the single frequency component is the largest. Meshing frequency component $f_{\mathrm{m}1}$ of the gear set S1 exists in the meshing force spectrum of the gear set S3 and will not disappear with the increase in rotational speed. As for the meshing forces of the

current gear set, the meshing frequency components of other gear sets only appear in part of the rotational speed, and the rotational speed of each gear set is different, indicating that the coupling relationship between the meshing forces of different gear sets is not equal.

To show the waterfall in more detail, meshing force spectrums of partial rotational speeds (1000 r/min, 2000 r/min, 4500 r/min, 7000 r/min) are shown in Figures 12–14. Figure 12 shows the meshing force spectrums of the first gear set. At the speeds of 1000 r/min and 2000 r/min, in addition to meshing frequency $f_{m1}$ and frequency doubling component of the first gear set, there is also meshing frequency $f_{m3}$ and frequency doubling component of the third gear set, which is caused by the connection between the sun gear of the third gear set and the inner ring gear of the first gear set. In addition, at 1000 r/min, there is the meshing frequency $f_{m2}$ component of the second gear set, which is caused by the parallel relationship between the first gear set and the second gear set. When the rotational speed rises to 4500 r/min and 7000 r/min, the meshing frequency components of other gear sets disappear, and only the meshing frequency and frequency doubling components of the first gear set exist.

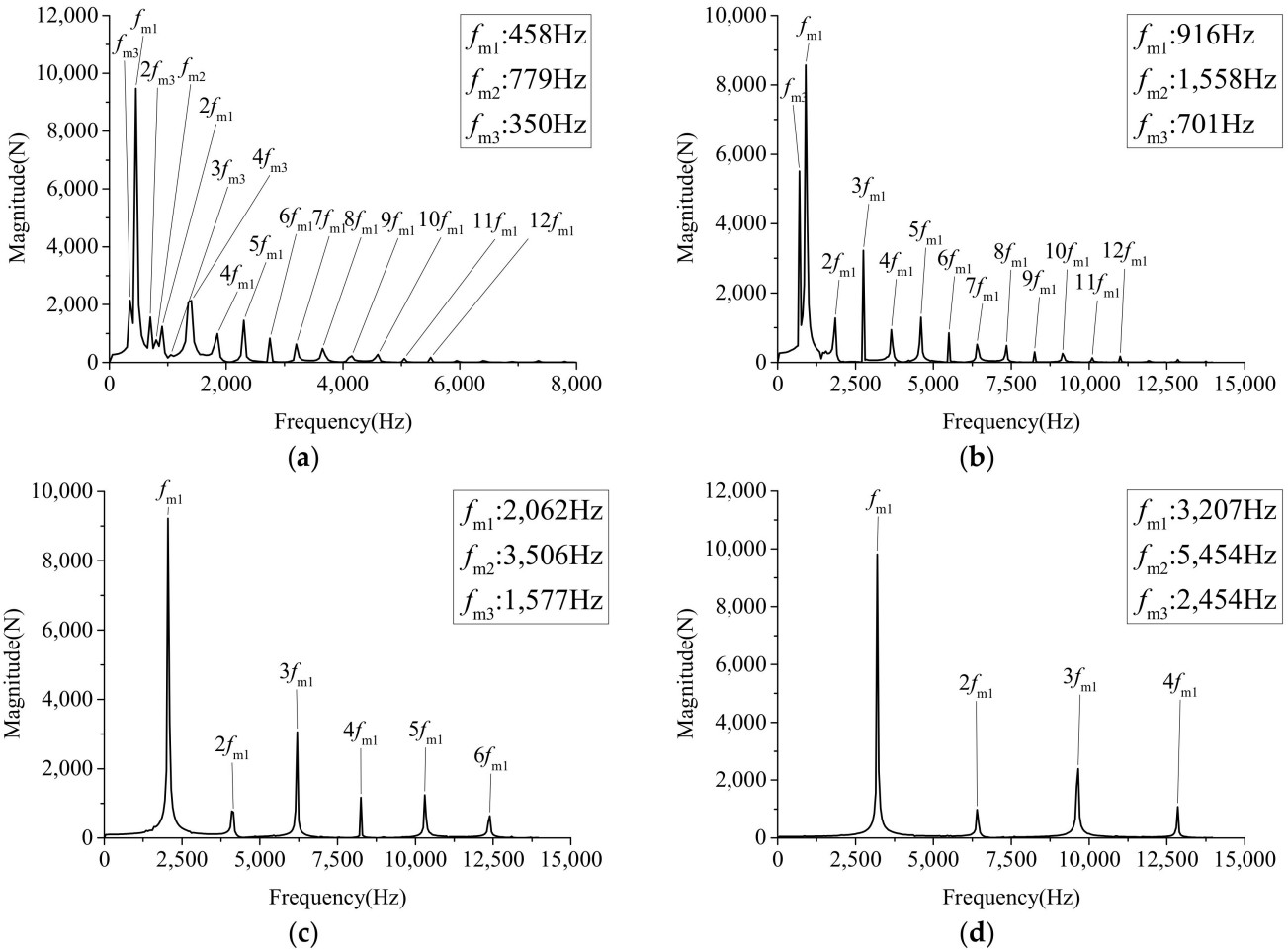

**Figure 12.** Spectrums diagram of meshing force of the first gear set at each input speed. (**a**) 1000 r/min; (**b**) 2000 r/min; (**c**) 4500 r/min; (**d**) 7000 r/min.

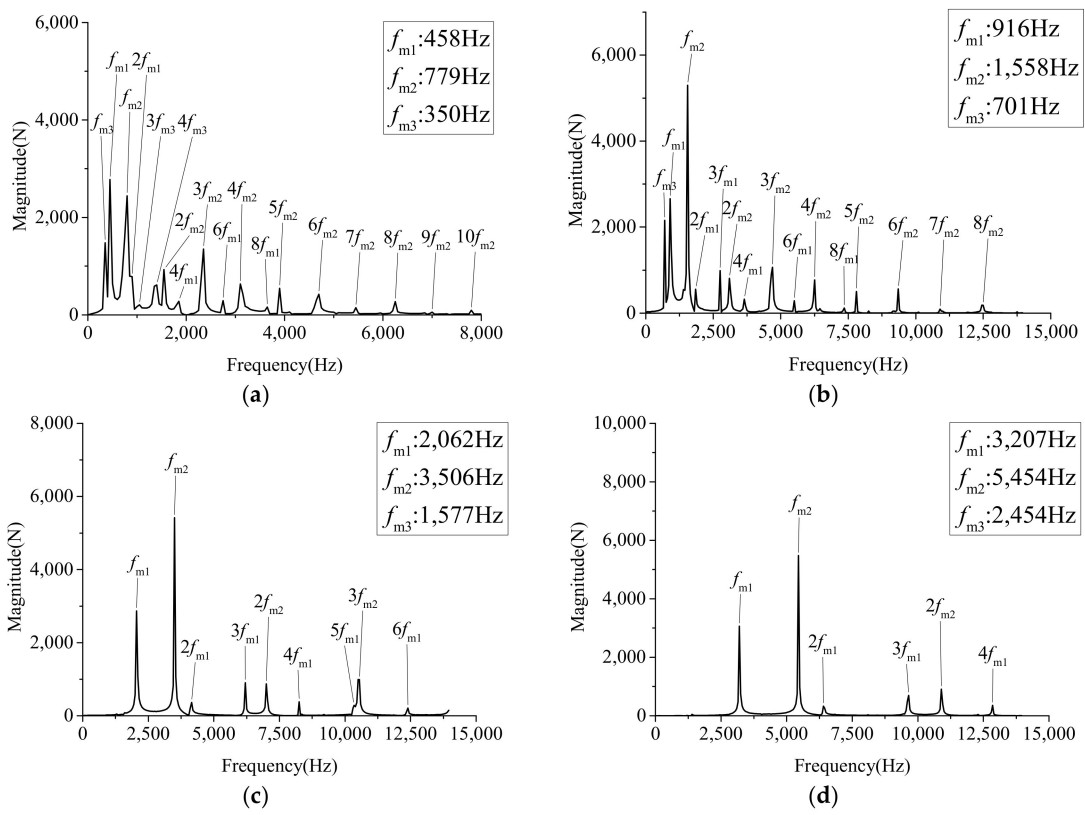

**Figure 13.** Spectrums diagram of meshing force of the second gear set at each input speed. (**a**) 1000 r/min; (**b**) 2000 r/min; (**c**) 4500 r/min; (**d**) 7000 r/min.

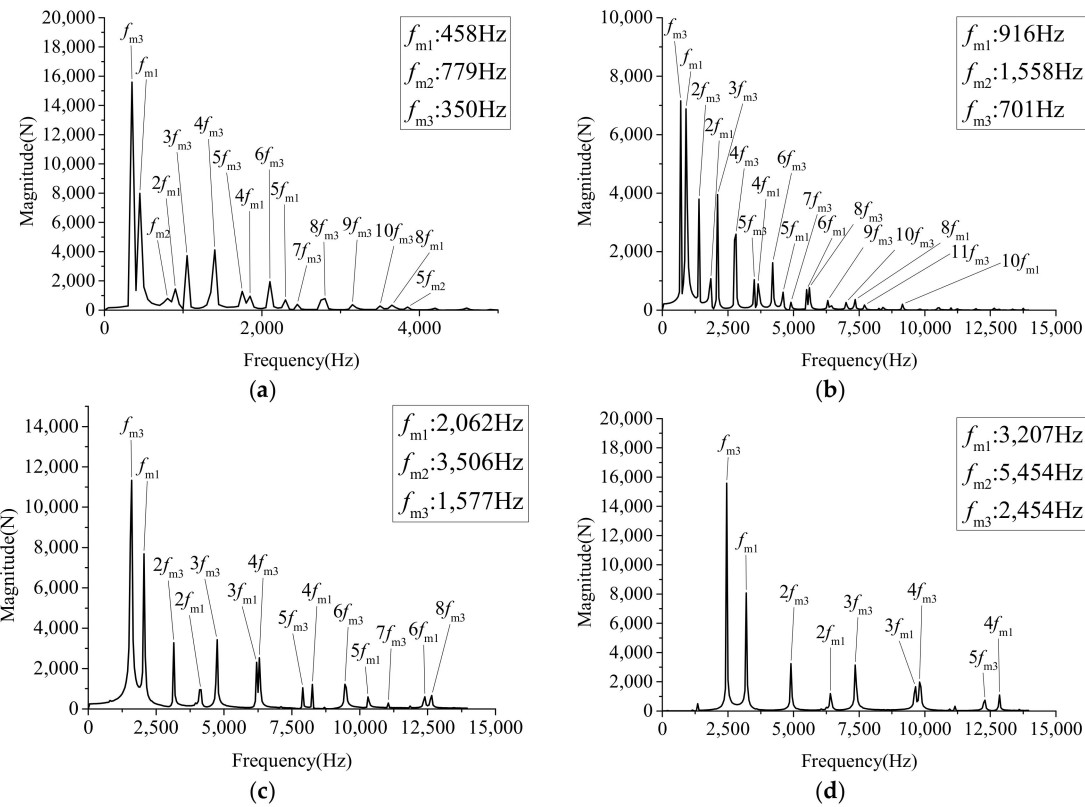

**Figure 14.** Spectrums diagram of meshing force of the third gear set at each input speed. (**a**) 1000 r/min; (**b**) 2000 r/min; (**c**) 4500 r/min; (**d**) 7000 r/min.

The meshing force spectrums of S2 in the second gear set is shown in Figure 13. As it has a parallel relationship with the first gear set and a series relationship with the third gear set in the overall structure, its composition is complicated. Moreover, the first and third gear set of single-fold meshing frequency components $f_{m1}$ and $f_{m3}$ and their corresponding frequency doubling components appeared, and even the first gear set of meshing frequency components $f_{m1}$ occupied the highest amplitude at 1000 r/min, indicating that the coupling effect of the parallel relation of the structure on the meshing force is more obvious at low rotational speed. When the speed increases to 4500 r/min and 7000 r/min, the meshing frequency component $f_{m3}$ of the third gear set S3 disappears, but the meshing frequency $f_{m1}$ and its frequency doubling component do not disappear with the increase in the speed.

Figure 14 shows the meshing force spectrums of S3 in the third gear set. Due to the connection between S3 in the third gear set and S1 in the first gear set, except the meshing frequency $f_{m3}$ and frequency doubling component of S3 in the third gear set, the meshing frequency $f_{m1}$ and frequency doubling component of S1 in the first gear set will not disappear with the increase in the speed at different input speeds. However, the amplitude of this component will continue to decrease with the increase in rotational speed. By observing the spectrum, it can be found that due to the characteristics of the overall structure, the coupling relationship between the meshing forces of different planetary rows is not symmetric, and the coupling phenomenon is more sensitive to low speeds.

## 4. Conclusions

Taking the three planetary gear sets transmission system as the research object, considering the characteristics of various parts, as well as the complex structure and changeable gear position of the planetary gear transmission system, a generalized dynamic model of the planetary gear set transmissions is established using the lump mass method, which can provide a reference for the subsequent automatic model establishment. This work contributes to the efficiency of dynamic modeling of multi-gear planets when they switch gears.

Through analysis of the variable speed of each frequency components of meshing force, the correctness of the above dynamic model is further verified. Analyzing the current gear set of meshing force, another gear set of meshing frequency components only appeared in the part of low rotational speed, and with the increase in rotational speed, some other low-frequency component in low speed could reduce or even disappear, but because of the characteristic of the structure, some other meshing frequency components will continue to exist, and will be unaffected by the speed change. In addition, the influence of different planetary gear sets on each other is strong or weak, indicating that the coupling relationship between meshing forces of different planetary gear sets is not equal; that is, the coupling relationship between meshing forces of different planetary gear sets is not symmetrical, and the coupling phenomenon of parallel row is more obvious. The results are similar to those of other literature. Resonance rotating speeds excited by meshing frequencies and their coupling frequencies are mainly concentrated in the range of middle–low rotating speed [20]. The frequency coupling of parallel row is more serious than that of series row [21].

In addition, in the establishment of a general dynamics modeling method, relevant matrix parameters still need to be manually input in the establishment process of a dynamic parameter gear matrix and shaft element connection, which can be combined with the development of related software in the subsequent correlation matrix of motion and torque analysis to complete the fully automated calculation and dynamics building process. Furthermore, based on linear time-varying models (LTVM), this paper focuses on the rapid dynamic model shifting method of a planetary gear sets system. Therefore, non-linear influencing factors such as gear backlash, impact and friction between gear pairs were ignored in the process of building the dynamic model in this paper, and the lack of persuasive analysis of dynamic characteristics gave this paper certain limitations in the subsequent analytical process. For spur gears, the impact of contact, disengagement, and re-contact between teeth caused by gear backlash mainly occurs in high-speed operation under light

load, and the analysis of the LTVM model is not sufficient. It is necessary to further analyze the dynamic characteristics of the system by adding non-linear factors and combining with the LTVM model.

**Author Contributions:** Conceptualization, H.W.; methodology, C.J.; software, C.J.; validation, C.J. and C.W.; formal analysis, H.W.; investigation, F.L.; resources, C.W. and F.L.; data curation X.S.; writing—original draft preparation, C.J.; writing—review and editing, H.W. All authors have read and agreed to the published version of the manuscript.

**Funding:** This research was funded by the National Key Laboratory of Science and Technology on Helicopter Transmission (Grant No. HTL-O-21G01).

**Conflicts of Interest:** The authors declare no conflict of interest.

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
