# Peer review of "A Generalized Dynamic Model and Coupling Meshing Force Analysis for Planetary Gear Set Transmissions"

_applsci, doi:10.3390/app12126279_

Round 1

Reviewer 1 Report

The article suits the scope of Applied Sciences. The subject refers to a permanent concern and challenge for research & development in the field of planetary gear drive systems. The work deals with the dynamic model of a three planetary gear set transmission, which is based on the lumped mass method. Such a theme is worthy of investigation, being one of great interest for the scientific community in the field. Several revisions are necessary for increasing the scientific value of the work and clarifying the approach, as indicated in the following points.

1. In the introductory section, the main objectives of the work should be clearly formulated in response to the critical analysis of the current research stage, going through the next path: the disadvantages of the existing methods ® the research opportunity ® the innovative approach; in the same idea, the innovative ideas should be better outlined.

2. In the background research, the authors should refer to their previous works in the field (if any), and highlight the elements that differentiate the approaches.

3. The background research and the references list as well should be extended with more recent works.

4. The research methodology should be more clearly described; a schematic representation of the research workflow could clarify the approach.

5. All parameters used in equations and figures should be explained with the first use; carefully check the whole paper in this regard.

6. In figures 9-11, use/write the full names of the intervening parameters and not just abbreviations (frequency instead of fre, magnitude instead of mag …); a similar remark for figures 12-14, with reference to magnitude.

7. The results in figures 9-14 should be commented on in more detail to better emphasize the usefulness of the proposed solution; the quality of these figures should also be improved.

8. The conclusions section should be extended & improved by a more detailed discussion on the research findings as well as on the future research directions, in terms of research opportunities opened by this work.

Author Response

Point 1: In the introductory section, the main objectives of the work should be clearly formulated in response to the critical analysis of the current research stage, going through the next path: the disadvantages of the existing methods ® the research opportunity ® the innovative approach; in the same idea, the innovative ideas should be better outlined.

Response 1: According to the different methods and factors considered in the establishment of system coupling dynamics model, the established dynamics model mainly includes the following types: lumped mass model, finite element model and lumped mass-finite element mixed model. However, the above modeling method can only establish the structure of components in a specific combination form. Aiming at the relationship between dynamic models established by different gears, the concept of gear matrix of dynamic parameter is introduced and relevant rules are formulated. A general dynamic model building method considering gears’ transformation is proposed. Thus, the dynamic model of the whole mechanism can be formed quickly, which is convenient for the subsequent dynamic characteristic analysis.

Point 2: In the background research, the authors should refer to their previous works in the field (if any), and highlight the elements that differentiate the approaches.

Response 2: For statics analysis of compound planetary gear sets, Kahraman[1] proposed a generalized formula for speed and force analysis of gear parts of planetary automatic transmission, which has certain applicability to compound planetary gear sets. In addition, researchers use graph theory to assist in kinematic and statical analysis. Lang[2] using graph theory to analyze the motion and torque of fixed gear train and epicyclic gear train. Lei[3] through the establishment of the corresponding matrix to assist to complete the planetary transmission motion and torque analysis. For dynamics modeling method, Chang[4] based on the idea of generalized finite element method (fem), the system discretization subunits of lumped mass model, proposed to the subunit assembly of the coupling system dynamic model of general modeling method. Based on this method, a general dynamic modeling method considering gears shift is proposed in this thesis.

  1. Kahraman, A.; Ligata, H.; Kienzle, K.; et al. A kinematics and power flow analysis methodology for automatic transmission planetary gear trains. Journal of Mechanical Design, 2004, 126(6), 1071-1081.
  2. Lang, S. Y. T. Graph-theoretic modelling of epicyclic gear systems. Mechanism and Machine Theory, 2005, 40(5), 511-529.
  3. Tian, L.; Li, Q. Matrix system for the analysis of planetary transmissions. Journal of Mechanical Design, 1997, 119(3), 333-337.
  4. Chang, L.; He, Z.; Liu, G. Dynamic modeling of parallel shaft gear transmissions using finite element method. Journal of Vibration and Shock, 2016, 35(20), 47-53.

Point 3: The background research and the references list as well should be extended with more recent works.

Response 3: The newly added literature is as follows:

  1. Chang, L.; He, Z.; Liu, G. Dynamic modeling of parallel shaft gear transmissions using finite element method. Journal of Vibration and Shock, 2016, 35(20), 47-53.
  2. Wei, J.; Zhang, A.; Qin, D.; et al. A coupling dynamics analysis method for a multistage planetary gear system. Mechanism and Machine Theory, 2017, 110, 27-49.
  3. Lv, H.; Li, Z.; Zhu, W.; et al. Construction of 12 DOFs spur gear coupling dynamic model. Vibroengineering PROCEDIA, 2019, 28, 241-245.

Point 4: The research methodology should be more clearly described; a schematic representation of the research workflow could clarify the approach.

Response 4: In this paper, appropriate supplements and amendments are made.

Point 5: All parameters used in equations and figures should be explained with the first use; carefully check the whole paper in this regard.

Response 5: The problem in formulas and graphs has been thoroughly checked.

Point 6: In figures 9-11, use/write the full names of the intervening parameters and not just abbreviations (frequency instead of fre, magnitude instead of mag…); a similar remark for figures 12-14, with reference to magnitude.

Response 6: The problem in figures 9-14 has been thoroughly checked.

Point 7: The results in figures 9-14 should be commented on in more detail to better emphasize the usefulness of the proposed solution; the quality of these figures should also be improved.

Response 7: Some comments has been added.

Point 8: The conclusions section should be extended & improved by a more detailed discussion on the research findings as well as on the future research directions, in terms of research opportunities opened by this work.

Response 8: In addition, in the establishment of general dynamics modeling method, relevant matrix parameters still need to be manually input in the establishment process of dynamic parameter gear matrix and shaft element connection, which can be combined with the development of related software in the subsequent correlation matrix of motion and torque analysis to complete the fully automated calculation and dynamics building process. In addition, there are many simplified parts in the process of model building in this paper, which can be further studied for the subsequent analysis of the dynamic characteristics of the structure.

Reviewer 2 Report

1. Overall assessment:

The authors deal with a linear dynamic model of a planetary gearbox. At the beginning, the model is widely commented. The evaluation of literature is well established with regard to the wide types of effects occurring in gears. However, it would be good if the subsequent model respects this initial appraisal. Nevertheless, the subsequent model is represented only by a linear self-adjoint system with an additive deterministic excitation, which neglects all these effects.

On the other hand, there are many useful models of this type and discrepancies are still emerging among them. The topic, therefore, deserves to be researched mainly at the level of a practical engineering platform and the results should be published in journals intended for engineering applications. APPLIED SCIENCES is a suitable platform for this type of articles.

2. Some remarks:

(a) Frankly speaking, it cannot be omitted that, there exist a vast number of papers published during the last 3-4 decades dealing with a sophisticated nonlinear formulation of gear mechanisms, see e.g. [4] and hundreds more. They discuss non-symmetric and one-sided constraints, respect various types of strong internal parametric excitations, internal dry friction, possible multiplicative random noises, deformability of elements and many other effects. All of them substantially influence dynamic stability of this system, energy throughput, reliability and lifetime period. See a lot of well known monographs.

(b) However, in the same time, the reviewer is aware that relatively simple models are needed as well. They enable a certain quick and very transparent qualitative assessment of gear mechanisms quality and feasibility. However, authors should carefully depict the domain of their model applicability. Therefore, a comprehensive paragraph should be necessarily included specifying the area of applicability and reasons why such a simple model is useful and applicable. Inaccessibility of relevant more complicated physical effect should be highlighted and well substantiated.

(c) Some terms introduced should be better explained and clearly quantified. For instance: e_0 – “comprehensive meshing error….“; \phi_{pi} – line 123, etc.

(d) All figures are too small, namely labels are mostly illegible.

(e)  English should be improved and checked by a native speaker. The meaning of many sentences can be only estimated and, therefore, they do not provide any clear information.

3. Conclusions:

The manuscript could be accepted and published in MDPI APPLIED SCIENCES Journal. However, the justification of the model should be clearly expressed and a domain of applicability depicted. Therefore, the re-review is necessary.

Author Response

Point 1: However, in the same time, the reviewer is aware that relatively simple models are needed as well. They enable a certain quick and very transparent qualitative assessment of gear mechanisms quality and feasibility. However, authors should carefully depict the domain of their model applicability. Therefore, a comprehensive paragraph should be necessarily included specifying the area of applicability and reasons why such a simple model is useful and applicable. Inaccessibility of relevant more complicated physical effect should be highlighted and well substantiated.

Response 1: Planetary row gear transmission system is the key shift device in the transmission, its performance directly affects the performance of the transmission. The device is mostly a coupling mechanism. Aiming at the different connection forms of the structure between different gears, the study on the generalization method suitable for different gears and the rapid composition of the dynamic equation of the whole mechanism can provide convenience for the subsequent dynamic characteristics analysis.

Point 2: Some terms introduced should be better explained and clearly quantified. For instance: e_0 – “comprehensive meshing error….“; \phi_{pi} – line 123, etc.

Response 2: This is my spelling mistake, it should be “composite meshing error”. And the existence of composite meshing error makes the actual meshing point of tooth surface deviate from the theoretical one, forming a displacement excitation in the meshing process.

Point 3: All figures are too small, namely labels are mostly illegible.

Response 3: The problem was checked and solved.

Point 4:   English should be improved and checked by a native speaker. The meaning of many sentences can be only estimated and, therefore, they do not provide any clear information.

Response 4: English sentences have been improved and checked by a native speaker.

Reviewer 3 Report

The paper is challenging to read because the statements are too long, and the main ideas are unclear. All the paper will benefit if a careful revision of the English is assured.

Introduction

Figure 1 is not appropriate in this section. Moreover, no reference is made relative to this figure in this section. The reference to this figure is made in section 2, the figure should be placed after it has been referenced.

Section 2

The authors do not show a correct procedure explaining how equation (2) appears.

Is the structural damping included in equation (15)? How is evaluated the damping matrix?

Results

How do these results compare with other published ones? 

Author Response

Point 1:

Introduction

Figure 1 is not appropriate in this section. Moreover, no reference is made relative to this figure in this section. The reference to this figure is made in section 2, the figure should be placed after it has been referenced.

Response 1: The figure has been replaced in the paper.

Point 2:

Section 2

The authors do not show a correct procedure explaining how equation (2) appears.

Is the structural damping included in equation (15)? How is evaluated the damping matrix?

Response 2: Equation (2) is a matrix form of differential equations of motion,the specific differential equations of motion is as follow:

where m1 and m2 respectively represent the lumped mass of node 1,2 connected by shaft element. I1 and I2 respectively represent the rotational inertia of node 1,2; kx and ky represent the bending stiffness of shaft element along X axis and Y axis respectively, and kθ represents the torsional stiffness of shaft element.Structural torsion damping Cθ and Bearing damping Cx.

,

In equation (15), The calculation process of structural damping is similar to the above equation.

Due to the limited space and the emphasis of this paper is not on this, only simplified matrix form is used in this paper.

Point 3:

Results

How do these results compare with other published ones?

Response 3: The results are similar to those of other literatures. Esonance rotating speeds excited by meshing frequencies and their coupling frequencies are mainly concentrated in the range of middle-low rotating speed[1]. The frequency coupling of parallel row is more serious than that of series row[2].

  1. Dou, Z.; Li, Y.; Zeng, Z.; et al. Frequency coupling and coupling resonance of a composite planetary transmission system under complex excitations. Journal of Vibration and Shock, 2019, 38(03), 16-23.
  2. Hao, X.; Yuan, X.; Bai, J.; et al. Analysis of factor affecting frequency coupling of double planetary gear system. Journal of Mechanical Transmission, 2020, 44(03), 110-117+165.

Round 2

Reviewer 2 Report

1. Authors verified the manuscript and amended a number of specific details. Indeed, I do not refuse a linear model, but there should be distinctly expressed, what are applicability limits of such a model (dynamic stability, reliability, parametric vibrations, etc.). However, the main recommendation concerning applicability of extremely simplified linear symmetric model remained without any reaction. Therefore, I repeat some of my previous comments:

“…Frankly speaking, it cannot be omitted that, there exist a vast number of papers published during the last 3-4 decades dealing with a sophisticated nonlinear formulation of gear mechanisms, see e.g. [4] and hundreds more. They discuss non-symmetric and one-sided constraints, respect various types of strong internal parametric excitations, internal dry friction, possible multiplicative random noises, deformability of elements and many other effects. All of them substantially influence dynamic stability of this system, energy throughput, reliability and lifetime period. See a lot of well known monographs….”.

Nothing of that is discussed in the new version of the manuscript.

2. I repeat also the remark regarding figures:

“…All figures are too small, namely labels are mostly illegible….“

No modifications in figures has been made.

Conclusion:

The paper could be published, but distinct real modifications in the main conception must be performed. Unfortunately, authors did not performed any modifications concerning the main objections formulated in the previous review until now. Therefore, I cannot recommend the actual version of the paper for acceptance and publication in the MDPI APPLIED SCIENCES.

Author Response

Point 1: Authors verified the manuscript and amended a number of specific details. Indeed, I do not refuse a linear model, but there should be distinctly expressed, what are applicability limits of such a model (dynamic stability, reliability, parametric vibrations, etc.). However, the main recommendation concerning applicability of extremely simplified linear symmetric model remained without any reaction. Therefore, I repeat some of my previous comments:

“…Frankly speaking, it cannot be omitted that, there exist a vast number of papers published during the last 3-4 decades dealing with a sophisticated nonlinear formulation of gear mechanisms, see e.g. [4] and hundreds more. They discuss non-symmetric and one-sided constraints, respect various types of strong internal parametric excitations, internal dry friction, possible multiplicative random noises, deformability of elements and many other effects. All of them substantially influence dynamic stability of this system, energy throughput, reliability and lifetime period. See a lot of well known monographs….”.

Nothing of that is discussed in the new version of the manuscript.

Response 1: Thanks for your comments, and we agree with you. At present, the linear model of gear system dynamics is very mature. However, the focus of this paper is to build an adaptive fast modeling method for the dynamic model of transmission gear transmission in multiple gears. We think this is the innovation of this paper. Based on Linear time-varying models (LTVM), this paper focuses on the rapid dynamic model shifting method of planetary gear sets system. Therefore, nonlinear influencing factors such as gear backlash, impact and friction between gear pairs were ignored in the process of building the dynamic model in this paper, and the lack of persuasive analysis of dynamic characteristics made this paper have certain limitations in the subsequent analysis process. For spur gears, the impact of contact, disengagement and re-contact between teeth caused by gear backlash mainly occurs in high-speed operation under light load; friction is the potential internal excitation source of the system, which will further aggravate the vibration of the system. So the analysis of LTVM model is not sufficient. It is necessary to further analyze the dynamic characteristics of the system by adding nonlinear factors and combining with LTVM model.

Point 2: I repeat also the remark regarding figures:

“…All figures are too small, namely labels are mostly illegible….“

No modifications in figures has been made.

Response 2: Sorry, we forgot to modify these contents in the last revised version. The labels in figures are updated. Please check these in the new revised version.

Reviewer 3 Report

the paper will benefit if some improvements will be presented as suggested by the reviewer. 

Author Response

Point 1: the paper will benefit if some improvements will be presented as suggested by the reviewer. 

Response 1: A number of specific details are amended. Please check these contents in the new revised version.

Round 3

Reviewer 2 Report

Even in the third version of the manuscript, the authors did not complete a clear definition of the range of problems, where their linear model is applicable. This area will not be large, because the key criterion for the usability of gear transmissions is always their dynamic stability and internal parametric processes (strongly nonlinear effects). Nevertheless, still doubtlessly there exists a certain area of applicability, but it must be precisely defined. On the other hand, the article contains a number of interesting steps that deserve publication. Therefore, although I did not find an agreement with authors on a certain compact version of the article, I recommend this one for publication, even if my reservations remain. For this reason, I do not want to delay the matter and thus cause a postponement of publication in the MDPI. Let the article be printed in its current form in the MDPI APPLIED SCIENCES.